# Quantification of Comfort for the Development of Binding Parts in a Standing Rehabilitation Robot

**DOI:** 10.3390/s23042206

**Published:** 2023-02-16

**Authors:** Yejin Nam, Sumin Yang, Jongman Kim, Bummo Koo, Sunghyuk Song, Youngho Kim

**Affiliations:** 1Department of Biomedical Engineering, Yonsei University, Wonju 26493, Republic of Korea; 2Department of Robotics & Mechatronics, Korea Institute of Machinery & Materials, Daejeon 34103, Republic of Korea

**Keywords:** rehabilitation robot, binding part, comfort, visual analog scale (VAS), interface pressure, tissue oxygen saturation (StO_2_), skin temperature

## Abstract

Human-machine interfaces (HMI) refer to the physical interaction between a user and rehabilitation robots. A persisting excessive load leads to soft tissue damage, such as pressure ulcers. Therefore, it is necessary to define a comfortable binding part for a rehabilitation robot with the subject in a standing posture. The purpose of this study was to quantify the comfort at the binding parts of the standing rehabilitation robot. In Experiment 1, cuff pressures of 10–40 kPa were applied to the thigh, shank, and knee of standing subjects, and the interface pressure and pain scale were obtained. In Experiment 2, cuff pressures of 10–20 kPa were applied to the thigh, and the tissue oxygen saturation and the skin temperature were measured. Questionnaire responses regarding comfort during compression were obtained from the subjects using the visual analog scale and the Likert scale. The greatest pain was perceived in the thigh. The musculoskeletal configuration affected the pressure distribution. The interface pressure distribution by the binding part showed higher pressure at the intermuscular septum. Tissue oxygen saturation (StO_2_) increased to 111.9 ± 6.7% when a cuff pressure of 10 kPa was applied and decreased to 92.2 ± 16.9% for a cuff pressure of 20 kPa. A skin temperature variation greater than 0.2 °C occurred in the compressed leg. These findings would help evaluate and improve the comfort of rehabilitation robots.

## 1. Introduction

Rehabilitation robots enable people to use immobile body parts and improve their functional independence. The development of robotic technology has increased the need for human–machine interfaces that provide user convenience [1,2]. The binding parts must be firmly fixed to the user for effective force transmission. Insufficient force may change the position of the binding parts or cause injury to the user. Prolonged, excessive loading can damage soft tissue, such as by causing pressure ulcers. Therefore, it is essential to maintain the various forces applied to the user within the safe limits of the systems [3]. Benson et al. evaluated the use of a commercial wearable robot in patients with chronic spinal cord injury (SCI), and at least 5 out of 10 subjects experienced skin abnormalities during a walking experiment [4]. It is difficult to determine the binding parts for SCI patients because they experience compressive tumors along with changes in skin architecture [5]. For the usability and acceptability of rehabilitation robots, it is important to quantify the physical interface at the binding part [6,7]. There are very few related studies conducted in the absence of clear guidelines and standardized approaches [8].

According to the International Association for the Study of Pain (IASP), pain is defined as “an unpleasant sensory and emotional experience expressed in relation to actual or potential tissue damage,” which is the most direct response to an excessive external load [9]. Several studies have been performed on musculoskeletal discomfort using tools such as the visual analog scale (VAS) [10,11,12], the Borg CR-10 scale [12], and localized muscle fatigue [13,14]. However, comfort measures can be influenced by various factors, such as friction, heat, muscle fatigue, and the effect on the movement, size, and weight of the device. Therefore, physiological signal analysis as a quantitative index is necessary to optimize the binding parts of rehabilitation robots.

A circumferential pneumatic cuff seems to be suitable for research in this field. A computerized cuff pressure algometry study showed that wider cuff widths and increased inflation rates were associated with faster pain perception [15]. The algometry is a useful tool for finding an acceptable interface pressure threshold; however, pain is a momentary response to an applied force, unlike the continuous external load generated by the wearable robot’s binding part [16]. A person requires a certain amount of time to react to a stimulus and, as a result, can withstand a greater force if the force is applied at a faster rate [17,18]. Therefore, comfort assessments should be measured under constant pressure and not for momentary pain.

The pathophysiological mechanisms underlying soft tissue injury have not yet been fully elucidated. Theoretically, focal ischemia, impaired lymphatic drainage, elevated local lactate levels, reperfusion injury, and persistent deformation of somatic cells are the primary causes, while the potential factors include malnutrition, age, medications, and circulatory disorders [19,20,21]. Normal and shear forces occur inside the soft tissue under a loading condition, and the relationship between the interface pressure and internal soft tissue stress is not linear. Its properties vary with thickness, tone, mechanical stiffness, and proximity to soft tissue [22,23,24]. The skin, adipose tissue, and muscle have different minimum pressures at which injury occurs, with muscles having the lowest threshold [20].

Many studies have been performed to determine the changes in blood flow to the skin and microvessels on applying tissue compression [25,26,27]. The thigh was more painful than the shank under the same pressure condition, and its oxygen saturation significantly increased [28]. This suggests that the thigh would be relatively responsive to physiological changes (oxygen deprivation) during tissue compression. Near-infrared spectroscopy has been used to monitor skeletal muscles, which are known to be most susceptible to pressure-induced injury. Near-infrared radiations with a wavelength of approximately 630–1300 nm have been often used to obtain blood information about arteries and veins by penetrating tissues to a depth of approximately 1–3 cm. By measuring changes in the concentrations of oxygenated hemoglobin (oxyHb) and deoxygenated hemoglobin (deoxyHb) obtained using the difference in absorbance at two wavelengths, neural activity in the relevant area could be measured continuously and noninvasively.

Many studies have been performed on the usability testing using rehabilitation robot prototypes; however, few have quantified comfort at the binding part. The purpose of this study is to quantify the comfort at the binding part of the standing rehabilitation robot. In Experiment 1, the comfort was compared for each binding part (thigh, shank, and knee). Anterior and posterior interface pressures were measured at the thigh and shank to confirm the pressure distribution according to the musculoskeletal configurations. In Experiment 2, tissue oxygen saturation and skin temperature were measured in the thigh during compression, since the thigh was most sensitive in the lower limb binding parts. Experiment 1 had a longer compression than Experiment 2. The questionnaire using VAS and Likert scales was used to determine comfort for every subject.

## 2. Materials and Methods

### 2.1. Participants

Forty healthy adults (30 males and 10 females) participated in this study. The participants had no neurological or musculoskeletal pathological findings in the lower extremities and had a body mass index (BMI) of less than 30 kg m^−2^. They were required to abstain from food intake 1 h prior to testing and from caffeine, nicotine, and alcohol 5 h prior to testing. They were asked to avoid strenuous exercise 24 h before the experiment. All participants were fully informed of the contents of the experiment and provided written consent to participate in this study. The experimental procedure was approved by the Institutional Review Board of Yovonseio University (1041849-202109-BM-152-03).

### 2.2. Questionnaire

The participants were asked to complete a questionnaire for comfort rating during compression. The questionnaire included a VAS with 0 (no pain) and 10 (the worst possible pain). The following questions were asked using a seven-point Likert scale.

Can you withstand that pressure for over an hour?Can you do your daily activities while maintaining that pressure?Are there any restrictions on movement?

### 2.3. Equipments

#### 2.3.1. A Pneumatic Cuff and a Compression Device

A pneumatic cuff (no-pinch single cuff, 86 cm × 10 cm (DTC-S07), DS Maref, Gunpo, Korea) was used. A preliminary experiment was performed using a cylindrical acrylic cylinder to confirm whether pressure was evenly applied in the circumferential direction. Pneumatic cuffs were worn on both legs. A compression device (MoorVMS-PRES; Moor Instruments, Axminster, UK) was used to apply pressure to the right leg of the pneumatic cuff. A cuff was worn on the left leg as a control group for changes in skin temperature. The pressure rise/fall were set as quickly as possible and completed within 10 s.

#### 2.3.2. Pressure Sensor

A mat-type pressure sensor (Pliance S2021, Novel GmbH, Munich, Germany) was used to measure the interface pressure data applied by the pneumatic cuff. The system consists of 16 × 16 pressure sensors capable of measuring up to 200 kPa; each cell size is 0.75 × 0.75 cm^2^. Interface pressures were measured at a 50 Hz sampling rate for 5 s under static pressure conditions.

The pressure distribution was determined using the average and standard deviation of the area, and the differences and characteristics of the interface pressure were confirmed. The position of the pressure sensor was confirmed via muscle palpation after the experiment. The pressure data were analyzed by subdividing the region of interest.

#### 2.3.3. Near-Infrared Spectroscopy (NIRS)

Near-infrared spectroscopy (MoorVMS-NIRS, Moor Instruments, Axminster, UK) was used to monitor changes in blood flow at the binding site. Near-infrared wavelengths of 750 and 850 nm were used, and the sampling rate was 5 Hz. NIRS can measure oxyHb and deoxyHb concentrations. TotalHb means the sum of oxyHb and deoxyHb, and the calculation of StO_2_ is as follows:(1)totalHb=oxyHb+deoxyHb
(2)$${\mathrm{StO}}_{2}=\frac{\mathrm{oxyHb}}{\mathrm{totalHb}}\times 100$$

The change in StO_2_ during compression was analyzed by setting an average of 1 min data stabilized in the standing baseline state to 100%.

#### 2.3.4. Temperature Sensor

Changes in the skin’s temperature were measured during compression using temperature sensors (RDXL6SD-USB, Omega Engineering, Norwalk, Canada), which were attached directly to the medial side of both legs. The sensitivity of the temperature sensor was 0.1 °C and was measured at a 1 Hz sampling rate during the experiment. The experiment was performed in a quiet room at a constant temperature (23 °C).

### 2.4. Experimental Protocol

#### 2.4.1. Experiment 1: Measurement of Pressure and Comfort for Each Binding Part

Twenty adults (10 males, 10 females, ages 24.8 ± 1.8 years old, 170.1 ± 8.3 cm, and 68.3 ± 11.4 kg) participated in the experiment to evaluate the comfort of each binding part. Pneumatic cuffs were worn on the right thigh, knee, and shank, which are primarily used as binding parts for wearable robots, and the comfort was evaluated by comparing the measured values during compression. The experiment was performed with the subjects standing upright. The pneumatic cuff was located at two-thirds of the height between the greater trochanter and the lateral epicondyle of the thigh. The pneumatic cuff was placed at the widest part of the shank, and the knee covered the patella. A pressure sensor was attached directly to the skin of the subject and placed over the pneumatic cuff. Measurements were conducted at five different locations (anterior/posterior thigh, anterior knee, and anterior/posterior shank). Four different pressures (10, 20, 30 and 40 kPa) were applied to the pneumatic cuff for 2 min. The interface pressure was measured for 5 s during compression. A 2 min rest was given at every pressure condition.

#### 2.4.2. Experiment 2: Measurement of Comfort and Physiological Changes in the Thigh

Fifteen males (23.1 ± 1.1 years old, 176.7 ± 3.7 cm, 75.5 ± 9.0 kg, BMI: 24.2 ± 2.5 kg m^−2^) participated in Experiment 2. An NIRS photoelectrode with an interaction distance of 50 mm was placed on the vastus lateralis at a two-thirds distance between the greater trochanter and the lateral epicondyle of the thigh (Figure 1a). The pressure sensor was placed on the back of the right thigh at the same height as the photoelectrode, and the temperature sensor was attached to the inner thigh so that the pneumatic cuff could cover all sensors. A pneumatic cuff was placed on the left thigh at the same height as the right thigh to check the skin temperature difference due to compression (Figure 1b). Each trial lasted for 5 min to reduce the effect of blood flow changes after standing. Cuff pressures of 10, 13.3, 16.7, and 20 kPa were applied for 5 min, respectively. The subject sat and rested for at least 5 min after each trial.

### 2.5. Statistical Analysis

One-way ANOVA and post-hoc Scheffé tests were used to determine the significant differences between the conditions. Pearson’s correlation analysis was performed to analyze the pain scale with the measured data. Statistical analyses were performed using IBM SPSS Statistics 25 (IBM, Armonk, NY, USA) (*p* < 0.05).

## 3. Results

### 3.1. Comfort and Interface Pressure at Binding Parts

#### 3.1.1. VAS

Figure 2 shows the pain scales for different cuff pressures at binding parts. The pain scale increased significantly (*p* < 0.05) with increasing cuff pressure for all binding parts. When a cuff pressure of 40 kPa was applied, the pain scales at the anterior and posterior of the thigh, the knee, and the anterior and posterior of the shank were 54.8 mm ± 16.0 mm, 55.2 mm ± 16.9 mm, 32.1 mm ± 14.2 mm, 32.4 mm ± 14.5 mm and 31.4 mm ± 16.2 mm, respectively. The pain scales were significantly larger in the thigh than in the shank or the knee (*p* < 0.05). When cuff pressures of 10, 20 and 30 kPa were applied, larger VAS values were observed in the anterior parts of the thigh than the posterior parts of the thigh; however, no significant difference existed.

#### 3.1.2. Masking of Interface Pressure Distributions

The pressure data were analyzed by masking regions of interest based on musculoskeletal structures. The location of the maximum pressure was confirmed by palpation. Among the whole mask (area: 16.88–20.25 cm^2^), three masks of the same area (8.44 cm^2^) were defined as medial, center, and lateral ones, as shown in Figure 3. Table 1 lists the interface pressure values for each mask for different cuff pressures at the binding parts. The mean interface pressures were not statistically different among the three masks, locating the rectus femoris muscle at the center mask in the anterior thigh. Alternatively, the center mask, which was located between the two muscles of the semitendinosus and biceps femoris in the posterior thigh, showed the highest interface pressure for all cuff pressures, which was statistically significant at the cuff pressure of 40 kPa (*p* < 0.05). For the knee, the medial part of the patella showed higher interface pressures, even though there was no statistical significance.

The tibia took the lowest interface pressures in the anterior part of the shank, which showed statistical significance at the cuff pressure of 20 kPa (*p* < 0.05). Conversely, the intermuscular septum between the medial and lateral gastrocnemius muscles showed the lowest interface pressures except for the cuff pressure of 10 kPa.

### 3.2. Comfort and Physiological Changes in the Thigh

#### 3.2.1. Questionnaire

The questionnaire responses for each cuff pressure are presented in Table 2. The results of the questionnaire using the Likert scale were calculated on a 1-point scale of “strongly disagree” and a 7-point scale of “strongly agree”. The pain scale significantly increased (*p* < 0.05) when the cuff pressure increased from 10 to 13.3 kPa. There was a significant difference in the pain scale with respect to different cuff pressures (F = 28.87, *p* < 0.05).

Most subjects responded positively to “Can you withstand for over an hour?” at cuff pressures of 10 and 13.3 kPa. At a cuff pressure of 16.7 kPa, five subjects answered “neutral,” and five answered negatively. At a cuff pressure of 20 kPa, twelve subjects responded negatively. Positive answers were reported to “Can you do your daily activities?” at 10 and 13.3 kPa in most subjects. However, negative answers prevailed at cuff pressures of 16.7 and 20 kPa. No restriction in movement was reported at cuff pressures of 10 and 13.3 kPa. However, nine subjects answered that the restriction occurred at a cuff pressure of 20 kPa.

#### 3.2.2. Interface Pressure

Table 3 shows the mean and standard deviation of the maximum pressure and its pressure gradient according to the cuff pressure. The maximum pressure significantly increased at all cuff pressures (*p* < 0.05). The pressure gradient according to the cuff pressure showed a statistically significant difference (F = 10.81, *p* < 0.05). There existed a strong correlation between the maximum pressure and its pressure gradient (*r* = 0.901, *p* < 0.05).

#### 3.2.3. Tissue Oxygen Saturation (StO_2_)

The average StO_2_ data for 1 min before compression was set as the baseline of 100%. Table 4 shows the StO_2_ for all subjects and the number of subjects in the groups in which the StO_2_ increased and decreased compared with the baseline and StO_2_ for each group. StO_2_ values for all subjects were measured at 111.9 ± 6.7%, 110.9 ± 5.9%, 107.6 ± 9.6%, and 92.2 ± 16.9% for cuff pressures of 10, 13.3, 16.7 and 20 kPa, respectively. At cuff pressures of 10 and 13.3 kPa, StO_2_ increased by 113.0 ± 5.6% and 112.3 ± 5.1% above baseline in fourteen subjects, respectively. StO_2_ increased to 112.1 ± 5.1% in thirteen subjects at 16.7 kPa, decreased by 89.0 ± 7.5% in two subjects, and decreased by 84.7 ± 13.4% in nine subjects at 20 kPa. StO_2_ gradually decreased below the baseline as the cuff pressure increased.

Figure 4 shows the individual StO_2_ changes for all the subjects with cuff pressure. In #4, #5, #9, #12, #14, and #15, StO_2_ increased at 13.3 kPa compared with 10 kPa and then decreased from 16.7 kPa. Subjects #2, #3, #6, #9, #13, and #14 maintained a similar level above the baseline until 13.3 kPa and then rapidly decreased below the baseline at 16.7 and 20 kPa. Subjects #1 and #11 increased above the baseline for all cuff pressures, which increased gradually. There was a correlation of −0.469 between the pain scale and StO_2_. For subjects whose StO_2_ decreased below baseline, it increased above baseline and returned to baseline immediately after the pressure was removed.

#### 3.2.4. Skin Temperature

As shown in Table 3, the maximum change in skin temperature of both legs was found at a cuff pressure of 10 kPa, compared with 13.3, 16.7, and 20 kPa. The mean skin temperature increased with cuff pressure in the compressed leg. The compressed leg had a higher mean skin temperature than the uncompressed leg for all cuff pressures. A temperature change of more than 0.2 °C occurred in the compressed leg. In the uncompressed leg, all subjects showed little change during compression.

## 4. Discussion

In this study, the interface pressure and pain scale were obtained by applying a cuff pressure between 10 and 40 kPa to each binding part (thigh, shank and knee). Physiological responses to StO_2_ and skin temperature were also measured in the thigh, where pain occurred the most.

In the preliminary experiments, when the pneumatic cuff was applied to the acrylic cylinder, the mean pressures were similar to the given cuff pressure, although slight deviations were found as the cuff flexed while wrapping around the curved surface. The measured normal force was 10% smaller than the applied force. It seems that the error resulted from the pressure threshold setting in the system and that the pressure is buffered due to the material properties of the pneumatic cuff. No statistical significance of the interface pressure according to the acrylic cylinder size was found (*p* > 0.05).

When pressure was applied to the thigh, shank, and knee, which are primarily used as binding parts for wearable robots, the most pain occurred in the thigh. A previous study reported that pain could be tolerated in areas with less soft tissue and that spatial summation of pain produced greater pain as muscle tissue volume increased [10]. This experiment confirmed a tendency for VAS to decrease as the subject’s BMI increased, although this was not statistically significant. Female subjects responded to a larger pain scale in response to cuff pressure than males, which contradicts the assumption that the adipose tissue acts as a cushion and can withstand higher pressure. A larger pain scale was obtained in the anterior thighs, which would be expected with more soft tissue present in the anterior thighs; however, there was no statistical significance. The thigh has a large amount of soft tissue, and the pressure data on the anterior thigh showed multiple patterns of pressure distribution depending on the subject’s biological sex and amount of tissue. Small interface pressure was found in the anterior shank, but large pressure was noted in the region between the bone and the soft tissue. This suggests that muscle thickness increases because of changes in muscle hardness and morphology when muscles are pressed against each other, and intermuscular pressure increases with the distance between muscle and tendon [29]. Therefore, greater pain in the thigh may have been caused by tissue compression due to the cuff inflation rather than just the interface pressure.

The experiment was performed by applying a cuff pressure of 10–20 kPa to the thigh for 5 min. The standing posture was maintained for 5 min to reduce the effects on blood flow and skin temperature due to the postural change from sitting to standing. Pain is a subjective sensation, and the pain scales reported by the subjects showed large deviations. Additional questions were asked using a Likert scale to obtain a detailed description of the subjects’ comfort. In the questionnaire results, slightly more positive answers were received for “Can you withstand for more than an hour?” than “can you do your daily activities?” In this study, the VAS at a cuff pressure of 13.3 kPa was 28.0 mm ± 11.2 mm, indicating that such pain could be tolerated in spite of discomfort in daily life. VAS would be better to quantify pain than the Likert scale, but its explanation might be a little ambiguous. Therefore, it would be more appropriate to quantify the comfort of rehabilitation robots under different conditions using both the VAS and the Likert scale [30].

A strong correlation existed between the maximal pressure value and the pain scale (*r* = 0.607). The pressure gradient is a critical indicator of tissue damage because pressure concentrations cause internal stress concentrations and soft tissue shear, resulting in high stress on soft tissues [31]. The pressure gradient at the maximum pressure showed a weak correlation (0.374) with the pain scale.

In Experiment 2, increases in StO_2_ over baseline were observed for cuff pressures of 10 and 13.3 kPa, which indicates that proper compression increases StO_2_ above baseline. However, excessive compression decreased StO_2_ below baseline, resulting from venous and arterial occlusion [32]. Thirteen out of 15 subjects showed more rapid decreases in StO_2_ at a cuff pressure of 16.7 kPa. At cuff pressures of 13.3 and 16.7 kPa, the mean value of StO_2_ increased to 110.9 ± 5.9% and 107.5 ± 9.6%, respectively, and decreased to 92.2 ± 16.9% from the baseline at a cuff pressure of 20 kPa. StO_2_ showed a correlation of −0.469 with the pain scale. A previous study performed in the supine position reported that arterial and venous occlusions occurred at pressures as low as 40–70 mmHg [33]. However, due to hemodynamic differences between standing and lying postures, higher pressures of 10 and 13.3 kPa (75 and 100 mmHg) did not cause vascular occlusion. The reduction in the blood vessel diameter of deep arrhythmias in the standing posture may result from isometric muscle contractions required to support the patient’s weight [34]. Therefore, further research would be required on blood flow changes in the thigh according to standing posture and various pressures. It would also be necessary to measure the muscle mass of the thigh and determine the correlation with blood flow due to isometric contraction of the erector muscle.

Additional experiments were performed in five subjects to confirm data changes over time. The same experimental protocol was followed by changing the pressure holding time to 10 min and requesting VAS at 1 min intervals from the subjects. In this study, VAS, StO_2_, and skin temperature were also measured at every 1 min interval during the 10 min of compression for five subjects. When a cuff pressure of 10 kPa was applied, no significant change in the pain scale was noted in four out of five subjects. StO_2_ increased over the baseline in all subjects. The skin temperature change increased the most compared with other cuff pressures. For a cuff pressure of 13.3 kPa, the pain scale increased over time, and StO_2_ became higher than the baseline in most subjects. The pain scale increased more rapidly over time at a cuff pressure of 16.7 kPa than at 10 and 13.3 kPa. When a cuff pressure of 16.7 kPa was applied, the pain scale increased over time more rapidly compared with when the cuff pressure was 10 or 13.3 kPa. The pain scales of two subjects (#1 and #4) increased and stabilized after 5 min. StO_2_ decreased below the baseline in two subjects, of which subject #2 stabilized after decreasing for 3 min. The skin temperature increased steadily in two subjects (#2 and #5) but was maintained in the remaining three subjects. At a cuff pressure of 20 kPa, the pain scale increased steadily in four subjects. The StO_2_ level decreased below the baseline in all subjects. In four out of five subjects, StO_2_ decreased rapidly upon compression and then stabilized after 4 min of compression.

Larger cuff pressure increased the pain scale over time. StO_2_ increased above the baseline at cuff pressures of 10 and 13.3 kPa and decreased below the baseline at cuff pressures of 16.7 and 20 kPa, but was stabilized within 5 min after the compression in most cases. The skin temperature increased during compression except for one subject (#1), but its change decreased with increasing cuff pressure. Table 5 shows the correlation coefficients between VAS and StO_2_ as well as skin temperature for each subject. There was a strong correlation between VAS and StO_2_, except for one subject (#5).

Despite the 10 min adaptation time for wearing the cuff, the largest change in skin temperature was found in both legs at an initial cuff pressure of 10 kPa. These results appear to have been caused by wearing the cuff and changes in systemic blood flow due to the compression. For other cuff pressures of 13.3 kPa, 16.7 kPa, and 20 kPa, the skin temperature of the compressed right leg showed a 0.2 °C or more increase in skin temperature compared with the uncompressed left leg. The skin temperature decreased in six of fifteen subjects when the cuff pressure was 20 kPa. There was a “negative” correlation of 0.472 between the skin temperature change in the compressed right leg and the pain scale. Previous studies have shown that temperature changes in the surrounding and local areas increased blood flow, and compression at the binding part affected the blood flow [35]. The correlation between StO_2_ and skin temperature was not statistically significant in the present study. According to previous research, exercise-induced StO_2_ did not decrease significantly at high skin temperature, making muscles less vulnerable to fatigue [36]. These results are related to temperature changes, skin temperature, blood flow, and chemical conditions. Temperature measurement is noninvasive, inexpensive, harmless to the human body, easy to use, and can be used for the first time to assess blood flow status and monitor hemodynamics [37]. Therefore, skin temperature changes should be investigated preferentially among the physiological changes in the binding region, and additional studies are required to determine the correlation with the change in blood flow.

This research proposes to evaluate the comfort of the binding part of a wearable robot; however, because the experiment was performed for only a part of the binding part, there is a possibility that there will be a difference in comfort. Although less pressure was applied to the thigh in Experiment 2 than in Experiment 1, a similar pain scale was reported. This may have varied across subject populations; however, there were also differences due to pressure and the maintenance of a standing posture over time. Some subjects showed an increase in VAS after 5 min of compression compared with the beginning of compression. These results indicate that VAS can increase with increasing compression time, even at the same pressure, and that additional experiments for long-term binding are required. As there is a significant deviation in comfort depending on the subject, it is necessary to conduct experiments with many subjects. It is challenging to accurately determine a subject’s muscle mass based on BMI alone. Measuring data at the binding site can result in the deformation of tissue geometry due to compression and other effects, such as perspiration at the measurement site. Post-occlusive reactive hyperemia might also occur when pressure is removed in the presence of lowered StO_2_.

## 5. Conclusions

This study aimed to quantify the comfort at the binding parts of a standing rehabilitation robot. In this study, VAS and Likert scales were used to analyze comfort according to the pressure of each binding part. To quantify the comfort, interface pressure, StO2, and skin temperature were measured. The most pain occurred in the thigh among the three parts (thigh, shank, and knee). Pain scales obtained from the subjects showed large deviations. It is appropriate to quantify rehabilitation robot comfort under different conditions using VAS and Likert scales. As a result of a questionnaire, a cuff pressure of 13.3 kPa or less seems to be a suitable pressure that does not affect daily life. The musculoskeletal composition affected the pressure distribution. Cuff pressures above 16.7 kPa seemed unsuitable for prolonged duration as StO2 decreased below the baseline. It can be seen that adverse effects on blood flow occurred in subjects whose skin temperature decreased by 20 kPa. These results suggested that pain is related to StO2 and skin temperature. There was a change in skin temperature for compression, even though no significant correlation with comfort was found. Further studies are needed to analyze the correlation between skin temperature and blood flow changes.

## Figures and Tables

**Figure 1 sensors-23-02206-f001:**
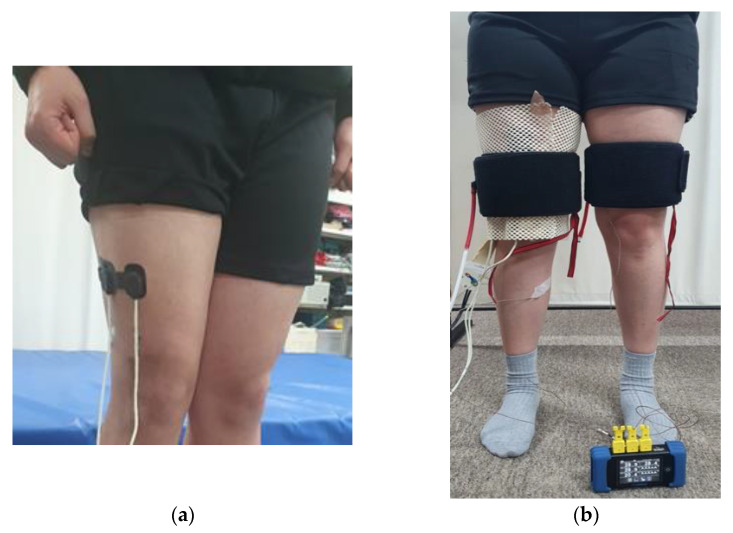
Equipment setting: (**a**) position of NIRS sensor probe; (**b**) experimental equipment attached to the thigh.

**Figure 2 sensors-23-02206-f002:**
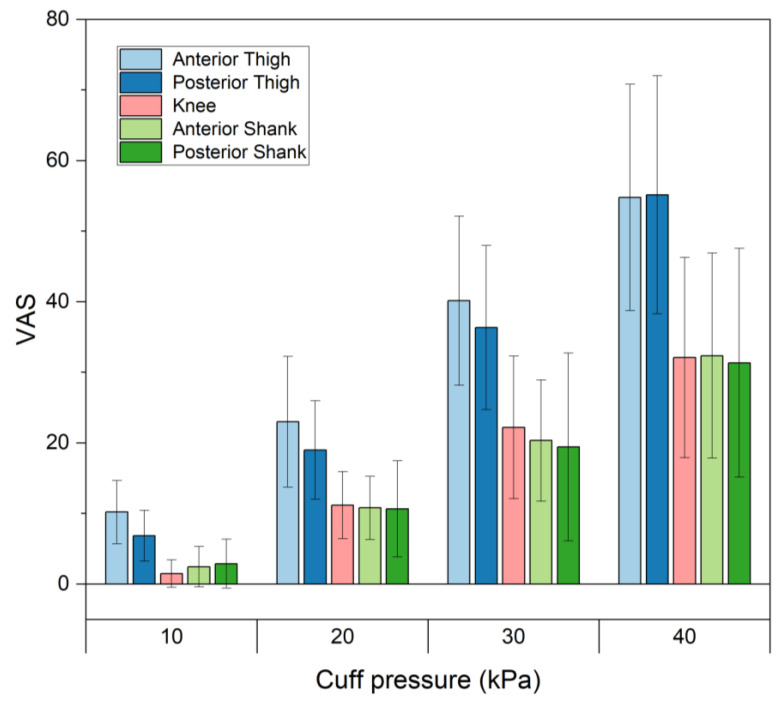
Pain scales for different cuff pressures at various binding parts.

**Figure 3 sensors-23-02206-f003:**
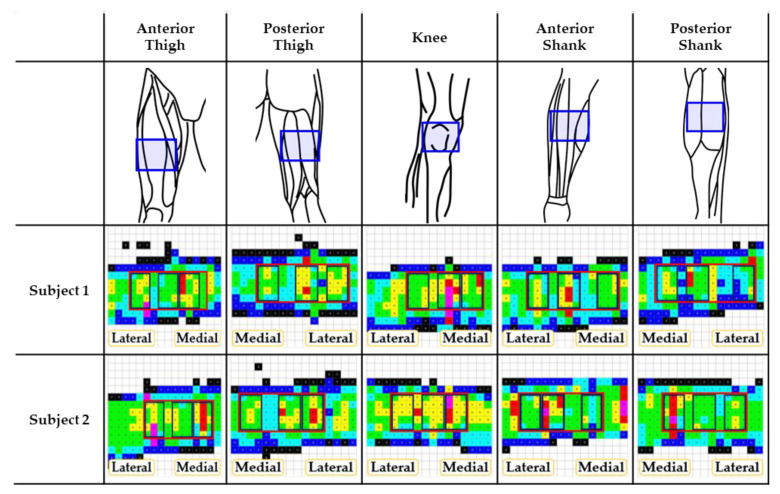
Musculoskeletal configurations of various binding parts and their pressure distributions. Pressure distributions correspond to the blue box in the 1^st^ row. The bottom two rows are examples of pressure distribution for 2 subjects out of 10 total. In the pressure distribution, red box represents the whole mask and sub-divided boxes represent medial, center, and lateral masks of interest.

**Figure 4 sensors-23-02206-f004:**
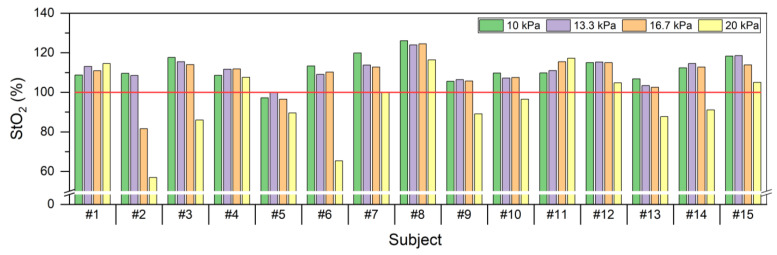
StO_2_ according to cuff pressures per subject in 15 subjects.

**Table 1 sensors-23-02206-t001:** Interface pressure for each mask.

	Cuff Pressure (kPa)	Interface Pressure for Each Mask (kPa)			
		Whole	Medial	Center	Lateral
Anterior Thigh	10	7.9 ± 0.9	7.8 ± 1.2	8.0 ± 1.6	8.2 ± 1.2
	20	18.1 ± 0.8	18.5 ± 1.8	18.3 ± 2.2	18.2 ± 1.7
	30	28.1 ± 1.1	28.9 ± 2.5	29.1 ± 3.5	28.1 ± 2.1
	40	37.5 ± 1.4	39.2 ± 2.9	38.2 ± 3.2	37.5 ± 2.6
Posterior Thigh	10	8.1 ± 0.8	8.1 ± 0.8	8.4 ± 1.5	7.9 ± 1.1
	20	17.9 ± 0.8	17.6 ± 1.6	18.5 ± 2.0	17.8 ± 1.7
	30	28.1 ± 1.0	27.3 ± 2.2	29.2 ± 3.1	28.3 ± 2.1
	40	38.1 ± 1.3	36.9 ± 2.6	39.5 ± 3.7	28.3 ± 3.3
Knee	10	7.8 ± 0.9	8.0 ± 1.0	8.0 ± 1.4	7.6 ± 1.0
	20	18.8 ± 1.3	19.3 ± 1.7	18.8 ± 2.4	18.9 ± 1.4
	30	29.7 ± 1.5	30.3 ± 1.9	29.8 ± 3.1	29.7 ± 2.0
	40	40.2 ± 1.6	40.7 ± 2.2	40.5 ± 3.3	40.1 ± 2.0
Anterior Shank	10	7.3 ± 0.7	7.3 ± 0.9	7.0 ± 1.0	7.6 ± 0.8
	20	17.4 ± 1.2	18.0 ± 1.7	16.3 ± 1.5	18.1 ± 1.2
	30	27.4 ± 1.3	28.6 ± 2.2	25.8 ± 1.8	28.5 ± 1.2
	40	37.6 ± 1.5	39.0 ± 2.5	35.9 ± 2.3	38.9 ± 1.3
Posterior Shank	10	7.3 ± 0.7	7.5 ± 1.0	7.4 ± 1.0	7.3 ± 0.7
	20	17.3 ± 0.8	17.6 ± 1.9	17.3 ± 1.9	17.5 ± 1.6
	30	27.5 ± 1.7	28.0 ± 3.0	27.7 ± 2.6	28.2 ± 2.6
	40	38.2 ± 1.5	38.5 ± 3.8	38.3 ± 3.0	38.9 ± 3.1

The shaded area represents the largest interface pressure among lateral, center, and medial masks.

**Table 2 sensors-23-02206-t002:** Questionnaire responses for each cuff pressure.

	Cuff Pressure (kPa)			
	10	13.3	16.7	20
VAS	15.2 ± 7.1	28.0 ± 11.2	42.4 ± 12.9	51.4 ± 13.7
Likert scale (1: strongly disagree, 7: strongly agree)				
Can you withstand for over an hour?	6.7 ± 0.9	5.7 ± 1.3	4.1 ± 1.4	2.9 ± 1.6
Can you do your daily activities while maintaining the pressure?	5.9 ± 1.0	4.9 ± 1.3	3.8 ± 1.3	2.5 ± 1.7
Are there any restrictions on movement?	1.6 ± 0.7	2.7 ± 0.9	3.7 ± 0.8	4.6 ± 1.1

**Table 3 sensors-23-02206-t003:** Mean ± SD of measured values according to the cuff pressure.

	Cuff Pressure (kPa)			
	10	13.3	16.7	20
Maximum pressure (kPa)	12.0 ± 2.2	19.0 ± 3.9	24.5 ± 4.3	31.6 ± 5.8
Pressure gradient (kPa/mm)	0.8 ± 0.3	1.2 ± 0.5	1.5 ± 0.6	2.1 ± 0.8
Skin temperature on the compressed leg (right) (°C)	32.2 ± 1.0	33.6 ± 0.8	33.8 ± 0.8	33.9 ± 0.8
Skin temperature on the unpressurized leg (left) (°C)	31.6 ± 0.7	32.6 ± 0.8	32.9 ± 0.6	32.8 ± 0.8
Temperature change on the compressed leg (right) (°C)	0.5 ± 0.3	0.2 ± 0.3	0.1 ± 0.3	0.0 ± 0.3
Temperature change on the unpressurized leg (left) (°C)	0.4 ± 0.1	0.1 ± 0.1	0.0 ± 0.2	0.0 ± 0.2

**Table 4 sensors-23-02206-t004:** Mean ± SD of StO_2_ for increased and decreased groups according to cuff pressure.

		Cuff Pressure (kPa)			
		10	13.3	16.7	20
**StO_2_ (%)**		111.9 ± 6.7	110.9 ± 5.9	107.5 ± 9.6	92.2 ± 16.9
Increase group	Number of subjects	14	14	13	6
	StO_2_ (%)	113.0 ± 5.6	112.3 ± 5.1	112.1 ± 5.1	110.9 ± 5.3
Decreased group	Number of subjects	1	1	2	9
	StO_2_ (%)	97.2	99.8	89.0 ± 7.5	84.7 ± 13.4

**Table 5 sensors-23-02206-t005:** Correlation coefficients between VAS and StO_2_ as well as skin temperature.

Subject	StO_2_	Skin Temperature
#1	−0.966	−0.527
#2	−0.844	0.986
#3	−0.907	0.897
#4	−0.921	0.580
#5	−0.354	0.346

## Data Availability

The data presented in this study are available upon request from the corresponding author. The data are not publicly available because the authors are continuing the study.

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
