# Peer review of "Quantification of Comfort for the Development of Binding Parts in a Standing Rehabilitation Robot"

_sensors, 2023, doi:10.3390/s23042206_

Round 1

Reviewer 1 Report

An overall good technical paper. But there are some concerns that might be addressed. 

1. equations are not cited in the text. 

2. more focus should be put on the design of experiments. I could not follow the overall experimentation process. 

3. The problem statement in the introduction section is not very clear. Please elaborate it. 

4. Conclusion should be point-wise. Commenting on each objective achieved. 

Author Response

Thanks for your comments for our manuscript. Based on your comments, the manuscript was revised. Response for each comment was attached in the file. 

Reviewer 2 Report

Paper needs improvement in terms of editing and analysis. To add value to the paper, author

should add contribution in the introduction section and give motivation of the study.

Further explore the current state of the art technologies and add some more background

to add more information in the paper. The following are the minor comments to be incorporated;

- Figures should be visible and well drawn.

- Captions of all figures should be explained very well.

- Conclusion should draw your achievement.

- Reference section should not be numbered.

The paper needs improvement in terms of editing and analysis. To add value to the paper, the author should contribute in the introduction section and give the motivation for the study. Further, explore the current state-of-the-art technologies and add more background to add more information to the paper. The following are the minor comments to be incorporated;

- Figures should be visible and well-drawn.

- Captions of all figures should be explained very well.

- Conclusion should draw your achievement. Compared with other research.

Conclusions should highlight your findings' insights and applicability; Please make them more concise and show only the high-impact outcomes. You need to explain the research novelty, limitations of the study, and contributions of the study for academics and practices, particularly for cleaner production and future work recommendation.

- Reference should include:

https://link.springer.com/article/10.1007/s11107-020-00914-8

Author Response

(The authors gave the same response as above.)

Round 2

Reviewer 1 Report

Paper can be accepted